# Meta-Learning Representations for Continual Learning

**Khurram Javed, Martha White**
Department of Computing Science
University of Alberta
T6G 1P8
kjaved@ualberta.ca, whitem@ualberta.ca

## Abstract

A continual learning agent should be able to build on top of existing knowledge to learn on new data quickly while minimizing forgetting. Current intelligent systems based on neural network function approximators arguably do the opposite—they are highly prone to forgetting and rarely trained to facilitate future learning. One reason for this poor behavior is that they learn from a representation that is not explicitly trained for these two goals. In this paper, we propose OML, an objective that directly minimizes catastrophic interference by learning representations that accelerate future learning and are robust to forgetting under online updates in continual learning. We show that it is possible to learn naturally sparse representations that are more effective for online updating. Moreover, our algorithm is complementary to existing continual learning strategies, such as MER and GEM. Finally, we demonstrate that a basic online updating strategy on representations learned by OML is competitive with rehearsal based methods for continual learning. [1]

## 1 Introduction

Continual learning—also called cumulative learning and lifelong learning—is the problem setting where an agent faces a continual stream of data, and must continually make and learn new predictions. The two main goals of continual learning are (1) to exploit existing knowledge of the world to quickly learn predictions on new samples (accelerate future learning) and (2) reduce interference in updates, particularly avoiding overwriting older knowledge. Humans, as intelligence agents, are capable of doing both. For instance, an experienced programmer can learn a new programming language significantly faster than someone who has never programmed before and does not need to forget the old language to learn the new one. Current state-of-the-art learning systems, on the other hand, struggle with both (French, 1999; Kirkpatrick et al., 2017).

Several methods have been proposed to address catastrophic interference. These can generally be categorized into methods that (1) modify the online update to retain knowledge, (2) replay or generate samples for more updates and (3) use semi-distributed representations. Knowledge retention methods prevent important weights from changing too much, by introducing a regularization term for each parameter weighted by its importance (Kirkpatrick et al., 2017; Aljundi et al., 2018; Zenke et al., 2017; Lee et al., 2017; Liu et al., 2018). Rehearsal methods interleave online updates with updates on samples from a model. Samples from a model can be obtained by replaying samples from older data (Lin, 1992; Mnih et al., 2015; Chaudhry et al., 2019; Riemer et al., 2019; Rebuffi et al., 2017; Lopez-Paz and Ranzato, 2017; Aljundi et al., 2019), by using a generative model learned on previous data (Sutton, 1990; Shin et al., 2017), or using knowledge distillation which generates targets using

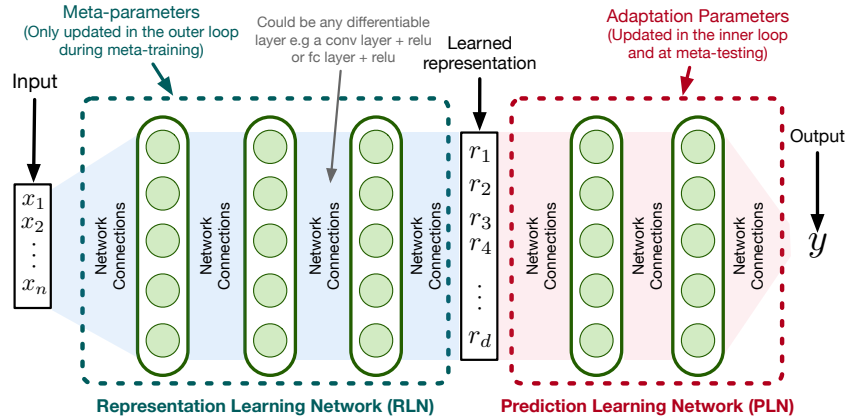

Figure 1: An example of our proposed architecture for learning representations for continual learning. During the inner gradient steps for computing the meta-objective, we only update the parameters in the prediction learning network (PLN). We then update both the representation learning network (RLN) and the prediction learning network (PLN) by taking a gradient step with respect to our meta-objective. The online updates for continual learning also only modify the PLN. Both RLN and PLN can be arbitrary models.

predictions from an older predictor (Li and Hoiem, 2018). These ideas are all complementary to that of learning representations that are suitable for online updating.

Early work on catastrophic interference focused on learning semi-distributed (also called sparse) representations (French, 1991, 1999). Recent work has revisited the utility of sparse representations for mitigating interference (Liu et al., 2019) and for using model capacity more conservatively to leave room for future learning (Aljundi et al., 2019). These methods, however, use sparsity as a proxy, which alone does not guarantee robustness to interference. A recently proposed online update for neural networks implicitly learns representations to obtain non-interfering updates (Riemer et al., 2019). Their objective maximizes the dot product between gradients computed for different samples. The idea is to encourage the network to reach an area in the parameter space where updates to the entire network have minimal interference and positive generalization. This idea is powerful: to specify an objective to explicitly mitigate interference—rather than implicitly with sparse representations.

In this work, we propose to explicitly learn a representation for continual learning that avoids interference and promotes future learning. We propose to train the representation with OML – a meta-objective that uses catastrophic interference as a training signal by directly optimizing through an online update. The goal is to learn a representation such that the stochastic online updates the agent will use at meta-test time improve the accuracy of its predictions in general. We show that using our objective, it is possible to learn representations that are more effective for online updating in sequential regression and classification problems. Moreover, these representations are naturally highly sparse. Finally, we show that existing continual learning strategies, like Meta Experience Replay (Riemer et al., 2019), can learn more effectively from these representations.

## 2  Problem Formulation

A Continual Learning Prediction (CLP) problem consists of an unending stream of samples

$$\mathcal{T} = (X_1, Y_1), (X_2, Y_2), \ldots, (X_t, Y_t), \ldots$$

for inputs $X_t$ and prediction targets $Y_t$, from sets $\mathcal{X}$ and $\mathcal{Y}$ respectively.[2] The random vector $Y_t$ is sampled according to an unknown distribution $p(Y|X_t)$. We assume the process $X_1, X_2, \ldots, X_t, \ldots$ has a marginal distribution $\mu : \mathcal{X} \to [0, \infty)$, that reflects how often each input is observed. This assumption allows for a variety of correlated sequences. For example, $X_t$ could be sampled from a distribution

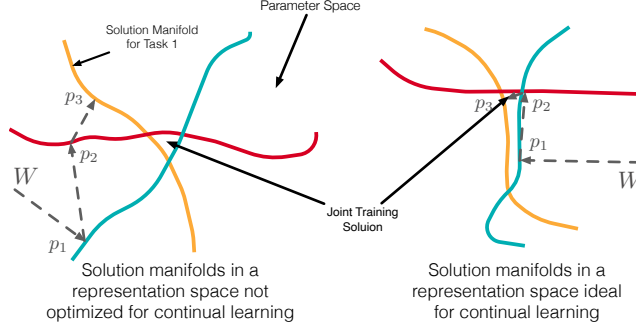

Figure 2: Effect of the representation on continual learning, for a problem where targets are generated from three different distributions $p_1(Y|x), p_2(Y|x)$ and $p_3(Y|x)$. The representation results in different solution manifolds for the three distributions; we depict two different possibilities here. We show the learning trajectory when training incrementally from data generates first by $p_1$, then $p_2$ and $p_3$. On the left, the online updates interfere, jumping between distant points on the manifolds. On the right, the online updates either generalize appropriately—for parallel manifolds—or avoid interference because manifolds are orthogonal.

potentially dependent on past variables $X_{t-1}$ and $X_{t-2}$. The targets $Y_t$, however, are dependent only on $X_t$, and not on past $X_i$. We define $\mathcal{S}_k = (X_{j+1}Y_{j+1}), (X_{j+2}Y_{j+2}) \ldots, (X_{j+k}, Y_{j+k})$, a random trajectory of length $k$ sampled from the CLP problem $\mathcal{T}$. Finally, $p(\mathcal{S}_k|\mathcal{T})$ gives a distribution over all trajectories of length $k$ that can be sampled from problem $\mathcal{T}$.

For a given CLP problem, our goal is to learn a function $f_{W,\theta}$ that can predict $Y_t$ given $X_t$. More concretely, let $\ell : \mathcal{Y} \times \mathcal{Y} \to \mathbb{R}$ be the function that defines loss between a prediction $\hat{y} \in \mathcal{Y}$ and target $y$ as $\ell(\hat{y}, y)$. If we assume that inputs $X$ are seen proportionally to some density $\mu : \mathcal{X} \to [0, \infty)$, then we want to minimize the following objective for a CLP problem:

$$\mathcal{L}_{CLP}(W, \theta) \stackrel{\text{def}}{=} \mathbb{E}[\ell(f_{W,\theta}(X), Y)] = \int \left[ \int \ell(f_{W,\theta}(x), y)p(y|x)dy \right] \mu(x)dx. \qquad (1)$$

where $W$ and $\theta$ represent the set of parameters that are updated to minimize the objective. To minimize $\mathcal{L}_{CLP}$, we limit ourselves to learning by online updates on a single $k$ length trajectory sampled from $p(\mathcal{S}_k|\mathcal{T})$. This changes the learning problem from the standard iid setting – the agent sees a single trajectory of correlated samples of length $k$, rather than getting to directly sample from $p(x, y) = p(y|x)\mu(x)$. This modification can cause significant issues when simply applying standard algorithms for the iid setting. Instead, we need to design algorithms that take this correlation into account.

A variety of continual problems can be represented by this formulation. One example is an online regression problem, such as predicting the next spatial location for a robot given the current location; another is the existing incremental classification benchmarks. The CLP formulation also allows for targets $Y_t$ that are dependent on a history of the most recent $m$ observations. This can be obtained by defining each $X_t$ to be the last $m$ observations. The overlap between $X_t$ and $X_{t-1}$ does not violate the assumptions on the correlated sequence of inputs. Finally, the prediction problem in reinforcement learning—predicting the value of a policy from a state—can be represented by considering the inputs $X_t$ to be states and the targets to be sampled returns or bootstrapped targets.

## 3 Meta-learning Representations for Continual Learning

Neural networks, trained end-to-end, are not effective at minimizing the CLP loss using a single trajectory sampled from $p(\mathcal{S}_k|\mathcal{T})$ for two reasons. First, they are extremely sample-inefficient, requiring multiple epochs of training to converge to reasonable solutions. Second, they suffer from catastrophic interference when learning online from a correlated stream of data (French, 1991). Meta-learning is effective at making neural networks more sample efficient (Finn et al., 2017). Recently, Nagabandi et al. (2019); Al-Shedivat et al. (2018) showed that it can also be used for quick adaptation from a stream of data. However, they do not look at the catastrophic interference problem. Moreover,

their work meta-learns a model initialization, an inductive bias we found insufficient for solving the catastrophic interference problem (See Appendix B.1).

To apply neural network to the CLP problem, we propose meta-learning a function $\phi_\theta(X)$ – a deep Representation Learning Network (RLN) parametrized by $\theta$ – from $\mathcal{X} \to \mathbb{R}^d$. We then learn another function $g_W$ from $\mathbb{R}^d \to \mathcal{Y}$, called a Prediction Learning Network (PLN). By composing the two functions we get $f_{W,\theta}(X) = g_W(\phi_\theta(X))$, which constitute our model for the CLP tasks as shown in Figure 1. We treat $\theta$ as meta-parameters that are learned by minimizing a meta-objective and then later fixed at meta-test time. After learning $\theta$, we learn $g_W$ from $\mathbb{R}^d \to \mathcal{Y}$ for a CLP problem from a single trajectory $\mathcal{S}$ using fully online SGD updates in a single pass. A similar idea has been proposed by Bengio et al. (2019) for learning causal structures.

For meta-training, we assume a distribution over CLP problems given by $p(\mathcal{T})$. We consider two meta-objectives for updating the meta-parameters $\theta$. (1) MAML-Rep, a MAML (Finn et al., 2017) like few-shot-learning objective that learns an RLN instead of model initialization, and OML (Online aware Meta-learning) – an objective that also minimizes interference in addition to maximizing fast adaptation for learning the RLN. Our OML objective is defined as:

$$\min_{W,\theta} \sum_{\mathcal{T}_i \sim p(\mathcal{T})} \text{OML}(W, \theta) \stackrel{\text{def}}{=} \sum_{\mathcal{T}_i \sim p(\mathcal{T})} \sum_{\mathcal{S}_k^j \sim p(\mathcal{S}_k|\mathcal{T}_i)} \left[ \mathcal{L}_{CLP_i}\Big( U(W, \theta, \mathcal{S}_k^j) \Big] \right. \tag{2}$$

where $S_k^j = (X_{j+1}^i Y_{j+1}^i), (X_{j+2}^i Y_{j+2}^i), \ldots, (X_{j+k}^i Y_{j+k}^i)$. $U(W_t, \theta, S_k^j) = (W_{t+k}, \theta)$ represents an update function where $W_{t+k}$ is the weight vector after $k$ steps of stochastic gradient descent. The $jth$ update step in $U$ is taken using parameters $(W_{t+j-1}, \theta)$ on sample $(X_{t+j}^i, Y_{t+j}^i)$ to give $(W_{t+j}, \theta)$.

MAML-Rep and OML objectives can be implemented as Algorithm 1 and 2 respectively, with the primary difference between the two highlighted in blue. Note that MAML-Rep uses the complete batch of data $\mathcal{S}_k$ to do $l$ inner updates – where $l$ is a hyper-parameter – whereas OML uses one data point from $\mathcal{S}_k$ for one update. This allows OML to take the effects of online continual learning – such as catastrophic forgetting – into account.

The goal of the OML objective is to learn representations suitable for online continual learnings. For an illustration of what would constitute an effective representation for continual learning, suppose that we have three clusters of inputs, which have significantly different $p(Y|x)$, corresponding to $p_1$, $p_2$ and $p_3$. For a fixed 2-dimensional representation $\phi_\theta : \mathcal{X} \to \mathbb{R}^2$, we can consider the manifold of solutions $W \in \mathbb{R}^2$ given by a linear model that pro-

---

**Algorithm 1:** Meta-Training : MAML-Rep

**Require:** $p(\mathcal{T})$: distribution over CLP problems
**Require:** $\alpha$, $\beta$: step size hyperparameters
**Require:** $l$: No of inner gradient steps
 1: randomly initialize $\theta$
 2: **while** not done **do**
 3:     randomly initialize $W$
 4:     Sample CLP problem $\mathcal{T}_i \sim p(\mathcal{T})$
 5:     Sample $\mathcal{S}_{train}$ from $p(\mathcal{S}_k|\mathcal{T}_i)$
 6:     $W_0 = W$
 7:     **for** $j$ in $1, 2, \ldots, l$ **do**
 8:         $W_j = W_{j-1} - \alpha \nabla_{W_{j-1}} \ell_i(f_{\theta,W_l}(S_{train}[:,0]), S_{train}[:,1])$
 9:     **end for**
10:     Sample $S_{test}$ from $p(\mathcal{S}_k|\mathcal{T}_i)$
11:     Update $\theta \leftarrow \theta - \beta \nabla_\theta \ell_i(f_{\theta,W_l}(S_{test}[:,0]), S_{test}[:,1])$
12: **end while**

---

vide equivalently accurate solutions for each $p_i$. These three manifolds are depicted as three different colored lines in the $W \in \mathbb{R}^2$ parameter space in Figure 2. The goal is to find one parameter vector $W$ that is effective for all three distributions by learning online on samples from three distributions sequentially. For two different representations, these manifolds, and their intersections can look very different. The intuition is that online updates from a $W$ are more effective when the manifolds are either parallel—allowing for positive generalization—or orthogonal—avoiding interference. It is unlikely that a representation producing such manifolds would emerge naturally. Instead, we will have to explicitly find it. By taking into account the effects of online continual learning, the OML objective optimizes for such a representation.

We can optimize this objective similarly to other gradient-based meta-learning objectives. Early work on learning-to-learn considered optimizing parameters through learning updates themselves, though typically considering approaches using genetic algorithms (Schmidhuber, 1987). Improvements

in automatic differentiation have made it more feasible to compute gradient-based meta-learning updates (Finn, 2018). Some meta-learning algorithms have similarly considered optimizations through multiple steps of updating for the few-shot learning setting (Finn et al., 2017; Li et al., 2017; Al-Shedivat et al., 2018; Nagabandi et al., 2019) for learning model initializations. The successes in these previous works in optimizing similar objectives motivate OML as a feasible objective for Meta-learning Representations for Continual Learning.

# 4  Evaluation

In this section, we investigate the question: can we learn a representation for continual learning that promotes future learning and reduces interference? We investigate this question by meta-learning the representations offline on a meta-training dataset. At meta-test time, we initialize the continual learner with this representation and measure prediction error as the agent learns the PLN online on a new set of CLP problems (See Figure 1).

---

**Algorithm 2:** Meta-Training : OML

**Require:** $p(\mathcal{T})$: distribution over CLP problems
**Require:** $\alpha$, $\beta$: step size hyperparameters
1: randomly initialize $\theta$
2: **while** not done **do**
3:     randomly initialize $W$
4:     Sample CLP problem $\mathcal{T}_i \sim p(\mathcal{T})$
5:     Sample $\mathcal{S}_{train}$ from $p(\mathcal{S}_k | \mathcal{T}_i)$
6:     $W_0 = W$
7:     **for** $j = 1, 2, \ldots, k$ **do**
8:         $(X_j, Y_j) = \mathcal{S}_{train}[j]$
9:         $W_j = W_{j-1} - \alpha \nabla_{W_{j-1}} \ell_i(f_{\theta, W_{j-1}}(X_j), Y_j)$
10:    **end for**
11:    Sample $S_{test}$ from $p(\mathcal{S}_k | \mathcal{T}_i)$
12:    Update $\theta \leftarrow \theta - \beta \nabla_\theta \ell_i(f_{\theta, W_k}(S_{test}[:, 0]), S_{test}[:, 1])$
13: **end while**

---

## 4.1  CLP Benchmarks

We evaluate on a simulated regression problem and a sequential classification problem using real data.

**Incremental Sine Waves:** An Incremental Sine Wave CLP problem is defined by ten (randomly generated) sine functions, with $x = (z, n)$ for $z \in [-5, 5]$ as input to the sine function and $n$ a one-hot vector for $\{1, \ldots, 10\}$ indicating which function to use. The targets are deterministic, where $(x, y)$ corresponds to $y = \sin_n(z)$. Each sine function is generated once by randomly selecting an amplitude in the range $[0.1, 5]$ and phase in $[0, \pi]$. A trajectory $\mathcal{S}_{400}$ from the CLP problem consists of 40 mini-batches from the first sine function in the sequence (Each mini-batch has eight elements), and then 40 from the second and so on. Such a trajectory has sufficient information to minimize loss for the complete CLP problem. We use a single regression head to predict all ten functions, where the input id $n$ makes it possible to differentiate outputs for the different functions. Though learnable, this input results in significant interference across different functions.

**Split-Omniglot:** Omniglot is a dataset of over 1623 characters from 50 different alphabets (Lake et al., 2015). Each character has 20 hand-written images. The dataset is divided into two parts. The first 963 classes constitute the meta-training dataset whereas the remaining 660 the meta-testing dataset. To define a CLP problem on this dataset, we sample an ordered set of 200 classes $(C_1, C_2, C_3, \ldots, C_{200})$. $\mathcal{X}$ and $\mathcal{Y}$, then, constitute of all images of these classes. A trajectory $\mathcal{S}_{1000}$ from such a problem is a trajectory of images – five images per class – where we see all five images of $C_1$ followed by five images of $C_2$ and so on. This makes $k = 5 \times 200 = 1000$. Note that the sampling operation defines a distribution $p(\mathcal{T})$ over problems that we use for meta-training.

## 4.2  Meta-Training Details

**Incremental Sine Waves:** We sample 400 functions to create our meta-training set and 500 for benchmarking the learned representation. We meta-train by sampling multiple CLP problems. During each meta-training step, we sample ten functions from our meta-training set and assign them task ids from one to ten. We concatenate 40 mini-batches – each with 32 x,y pairs – generated from function one, then function two and so on, to create our training trajectory $\mathcal{S}_{400}$. For evaluation, we similarly randomly sample ten functions from the test set and create a single trajectory. We use SGD on the MSE loss with a mini-batch size of 8 for online updates, and Adam (Kingma and Ba, 2014) for optimizing the OML objective. Note that the OML objective involves computing gradients through

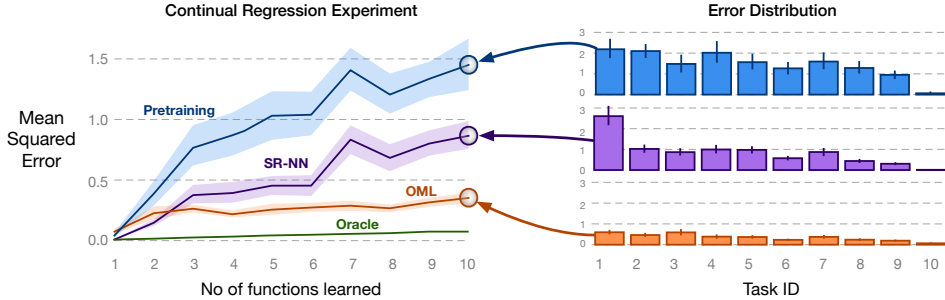

Figure 3: Mean squared error across all 10 regression tasks. The x-axis in (a) corresponds to seeing all data points of samples for class 1, then class 2 and so on. These learning curves are averaged over 50 runs, with error bars representing 95% confidence interval drawn by 1,000 bootstraps. We can see that the representation trained on iid data—Pre-training—is not effective for online updating. Notice that in the final prediction accuracy in (b), Pre-training and SR-NN representations have accurate predictions for task 10, but high error for earlier tasks. OML, on the other hand, has a slight skew in error towards later tasks in learning but is largely robust. Oracle uses iid sampling and multiple epochs and serves as a best case bound.

a network unrolled for 400 steps. At evaluation time, we use the same learning rate as used during the inner updates in the meta-training phase for OML. For our baselines, we do a grid search over learning rates and report the results for the best performing parameter.

We found that having a deeper representation learning network (RLN) improved performance. We use six layers for the RLN and two layers for the PLN. Each hidden layer has a width of 300. The RLN is only updated with the meta-update and acts as a fixed feature extractor during the inner updates in the meta-learning objective and at evaluation time.

**Split-Omniglot:** We learn an encoder – a deep CNN with 6 convolution and two FC layers – using the MAML-Rep and the OML objective. We treat the convolution parameters as $\theta$ and FC layer parameters as $W$. Because optimizing the OML objective is computationally expensive for $H = 1000$ (It involves unrolling the computation graph for 1,000 steps), we approximate the two objectives. For MAML-Rep we learn the $\phi_\theta$ by maximizing fast adaptation for a 5 shot 5-way classifier. For OML, instead of doing $|\mathcal{S}_{train}|$ no of inner-gradient steps as described in Algorithm 2, we go over $\mathcal{S}_{train}$ five steps at a time. For $kth$ five steps in the inner loop, we accumulate our meta-loss on $\mathcal{S}_{test}[0 : 5 \times k]$, and update our meta-parameters using these accumulated gradients at the end as explained in Algorithm 3 in the Appendix. This allows us to never unroll our computation graphs for more than five steps (Similar to truncated back-propagation through time) and still take into account the effects of interference at meta-training.

Finally, both MAML-Rep and OML use 5 inner gradient steps and similar network architectures for a fair comparison. Moreover, for both methods, we try multiple values for the inner learning rate $\alpha$ and report the results for the best parameter. For more details about hyper-parameters see the Appendix. For more details on implementation, see Appendix A.

### 4.3 Baselines

We compare MAML-Rep and OML – the two Meta-learneing based Representations Leanring methods to three baselines.

**Scratch** simply learns online from a random network initialization, with no meta-training.

**Pre-training** uses standard gradient descent to minimize prediction error on the meta-training set. We then fix the first few layers in online training. Rather than restricting to the same 6-2 architecture for the RLN and PLN, we pick the best split using a validation set.

**SR-NN** use the Set-KL method to learn a sparse representation (Liu et al., 2019) on the meta-training set. We use multiple values of the hyper-parameter $\beta$ for SR-NN and report results for one that performs the best. We include this baseline to compare to a method that learns a sparse representation.

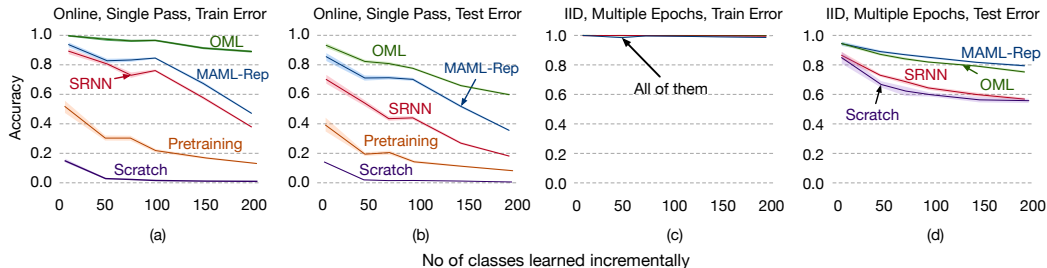

Figure 4: Comparison of representations learned by the MAML-Rep, OML objective and the baselines on Split-Omniglot. All curves are averaged over 50 CLP runs with 95% confidence intervals drawn using 1,000 bootstraps. At every point on the x-axis, we only report accuracy on the classes seen so far. Even though both MAML-Rep and OML learn representations that result in comparable performance of classifiers trained under the IID setting (c and d), OML out-performs MAML-Rep when learning online on a highly correlated stream of data showing it learns representations more robust to interference. SR-NN, which does not do meta-learning, performs worse even under the IID setting showing it learns worse representations.

## 4.4 Meta-Testing

We report results of $\mathcal{L}_{CLP}(W_{online}, \theta_{meta})$ for fully online updates on a single $\mathcal{S}_k$ for each CLP problem. For each of the methods, we separately tune the learning rate on a five validation trajectories and report results for the best performing parameter.

**Incremental Sine Waves:** We plot the average mean squared error over 50 runs on the full testing set, when learning online on unseen sequences of functions, in Figure 3 (left). OML can learn new functions with a negligible increase in average MSE. The Pre-training baseline, on the other hand, clearly suffers from interference, with increasing error as it tries to learn more and more functions. SR-NN, with its sparse representation, also suffers from noticeably more interference than OML. From the distribution of errors for each method on the ten functions, shown in Figure 3 (right), we can see that both Pre-training and SR-NN have high errors for functions learned in the beginning whereas OML performs only slightly worse on those.

**Split-Omniglot:**

We report classification accuracy on the training trajectory ($\mathcal{S}_{train}$) as well as the test set in Figure 4. Note that training accuracy is a meaningful metric in continual learning as it measures forgetting. The test set accuracy reflects both forgetting and generalization error. Our method can learn the training trajectory almost perfectly with minimal forgetting. The baselines, on the other hand, suffer from forgetting as they learn more classes sequentially. The higher training accuracy of our method also translates into better generalization on the test set. The difference in the train and test performance is mainly due to how few samples are given per class: only 15 for training and 5 for testing.

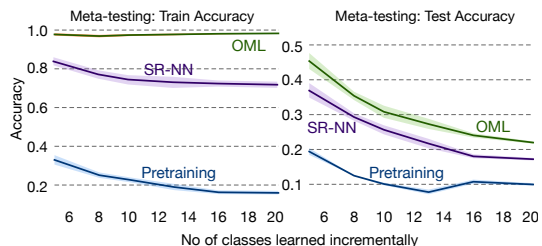

Figure 5: OML scales to more complex datasets such a Mini-imagenet. We use the existing meta-training/meta-testing split of mini-imagenet. At meta-testing, we learn a 20 way classifier using 30 samples per class.

As a sanity check, we also trained classifiers by sampling data IID for 5 epochs and report the results in Fig. 4 (c) and (d). The fact that OML and MAML-Rep do equally well with IID sampling indicates that the quality of representations ($\phi_\theta = \mathbb{R}^d$) learned by both objectives are comparable and the higher performance of OML is indeed because the representations are more suitable for incremental learning.

Moreover, to test if OML can learn representations on more complex datasets, we run the same experiments on mini-imagenet and report the results in Figure 5.

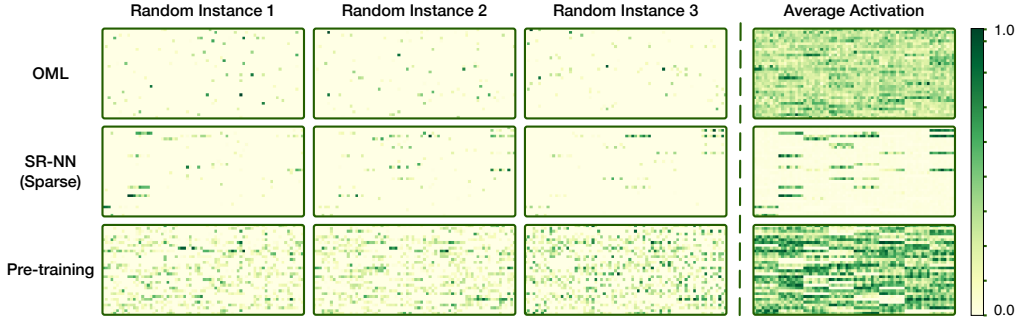

Figure 6: We reshape the 2304 length representation vectors into 32x72, normalize them to have a maximum value of one and visualize them; here random instance means representation for a randomly chosen input from the training set, whereas average activation is the mean representation for the complete dataset. For SR-NN, we re-train the network with a different value of parameter $\beta$ to have the same instance sparsity as OML. Note that SR-NN achieves this sparsity by never using a big part of representation space. OML, on the other hand, uses the full representation space. In-fact, OML has no dead neurons whereas even pre-training results in some part of the representation never being used.

## 4.5 What kind of representations does OML learn?

As discussed earlier, French (1991) proposed that sparse representations could mitigate forgetting. Ideally, such a representation is instance sparse–using a small percentage of activations to represent an input– while also utilizing the representation to its fullest. This means that while most neurons would be inactive for a given input, every neuron would participate in representing some input. Dead neurons, which are inactive for all inputs, are undesirable and may as well be discarded. An instance sparse representation with no dead neurons reduces forgetting because each update changes only a small number of weights which in turn should only affect a small number of inputs. We hypothesize that the representation learned by OML will be sparse, even though the objective does not explicitly encourage this property.

We compute the average instance sparsity on the Omniglot training set, for OML, SR-NN, and Pre-training. OML produces the most sparse network, without any dead neurons. The network learned by Pre-training, in comparison, uses over 10 times more neurons on average to represent an input. The best performing SR-NN used in Figure 4 uses 4 times more neurons. We also re-trained SR-NN with a parameter to achieve a similar level of sparsity as OML, to compare representations of similar sparsity rather than representations chosen based on accuracy. We use $\beta = 0.05$ which results in an instance sparsity similar to OML.

We visualize all the solutions in Figure 6. The plots highlight that OML learns a highly sparse and well-distributed representation, taking the most advantage of the large capacity of the representation. Surprisingly, OML has no dead neurons, which is a well-known problem when learning sparse representations (Liu et al., 2019). Even Pre-training, which does not have an explicit penalty to enforce sparsity, has some dead neurons. Instance sparsity and dead neurons percentage for each method are reported in Table 1.

Table 1: Instance sparisty and dead neuron percentage for different methods. OML learns highly sparse representations without any dead neurons. Even Pre-training, which does not optimize for sparsity, ends up with some dead neurons, on the other hand.

| Method | Instance Sparsity | Dead Neurons |
|---|---|---|
| OML | 3.8% | 0% |
| SR-NN (Best) | 15% | 0.7% |
| SR-NN (Sparse) | 4.9% | 14% |
| Pre-Training | 38% | 3% |

## 5 Improvements by Combining with Knowledge Retention Approaches

We have shown that OML learns effective representations for continual learning. In this section, we answer a different question: how does OML behave when it is combined with existing continual

Table 2: OML combined with existing continual learning methods. All memory-based methods use a buffer of 200. Error margins represent one std over 10 runs. Performance of all methods is considerably improved when they learn from representations learned by OML moreover, even online updates are competitive with rehearsal based methods with OML. Finally, online updates on OML outperform all methods when they learn from other representations. Note that MER does better than approx IID in some cases because it does multiple rehearsal-based updates for every sample.

| | Split-Omniglot | | | | | |
| | One class per task, 50 tasks | | | Five classes per task, 20 tasks | | |
| Method | Standard | OML | Pre-training | Standard | OML | Pre-training |
|---|---|---|---|---|---|---|
| Online | $04.64_{\pm 2.61}$ | $\mathbf{64.72}_{\pm 2.57}$ | $21.16_{\pm 2.71}$ | $01.40_{\pm 0.43}$ | $\mathbf{55.32}_{\pm 2.25}$ | $11.80_{\pm 1.92}$ |
| Approx IID | $53.95_{\pm 5.50}$ | $\mathbf{75.12}_{\pm 3.24}$ | $54.29_{\pm 3.48}$ | $48.02_{\pm 5.67}$ | $\mathbf{67.03}_{\pm 2.10}$ | $46.02_{\pm 2.83}$ |
| ER-Reservoir | $52.56_{\pm 2.12}$ | $\mathbf{68.16}_{\pm 3.12}$ | $36.72_{\pm 3.06}$ | $24.32_{\pm 5.37}$ | $\mathbf{60.92}_{\pm 2.41}$ | $37.44_{\pm 1.67}$ |
| MER | $54.88_{\pm 4.12}$ | $\mathbf{76.00}_{\pm 2.07}$ | $62.76_{\pm 2.16}$ | $29.02_{\pm 4.01}$ | $\mathbf{62.05}_{\pm 2.19}$ | $42.05_{\pm 3.71}$ |
| EWC | $05.08_{\pm 2.47}$ | $\mathbf{64.44}_{\pm 3.13}$ | $18.72_{\pm 3.97}$ | $02.04_{\pm 0.35}$ | $\mathbf{56.03}_{\pm 3.20}$ | $10.03_{\pm 1.53}$ |

learning methods? We test the performance of EWC (Kirkpatrick et al., 2017), MER (Riemer et al., 2019) and ER-Reservoir (Chaudhry et al., 2019), in their standard form—learning the whole network online—as well as with pre-trained fixed representations. We use pre-trained representations from OML and Pre-training, obtained in the same way as described in earlier sections. For the Standard online form of these algorithms, to avoid the unfair advantage of meta-training, we initialize the networks by learning iid on the meta-training set.

As baselines, we also report results for (a) fully online SGD updates that update one point at a time in order on the trajectory and (b) approximate IID training where SGD updates are used on a random shuffling of the trajectory, removing the correlation.

We report the test set results for learning 50 tasks with one class per task and learning 20 tasks with 5 tasks per class in Split-Omniglot in Table 2. For each of the methods, we do a 15/5 train/test split for each Omniglot class and test multiple values for all the hyperparameters and report results for the best setting. The conclusions are surprisingly clear. (1) OML improves all the algorithms; (2) simply providing a fixed representation, as in Pre-training, does not provide nearly the same gains as OML and (3) OML with a basic Online updating strategy is already competitive, outperforming all the continual learning methods without OML. There are a few additional outcomes of note. OML outperforms even approximate IID sampling, suggesting it is not only mitigating interference but also making learning faster on new data. Finally, the difference in online and experience replay based algorithms for OML is not as pronounced as it is for other representations.

## 6 Conclusion

In this paper, we proposed a meta-learning objective to learn representations that are robust to interference under online updates and promote future learning. We showed that using our representations, it is possible to learn from highly correlated data streams with significantly improved robustness to forgetting. We found sparsity emerges as a property of our learned representations, without explicitly training for sparsity. We finally showed that our method is complementary to the existing state of the art continual learning methods, and can be combined with them to achieve significant improvements over each approach alone.

An important next step for this work is to demonstrate how to learn these representations online without a separate meta-training phase. Initial experiments suggest it is effective to periodically optimize the representation on a recent buffer of data, and then continue online update with this updated fixed representation. This matches common paradigms in continual learning—based on the ideas of a sleep phase and background planning—and is a plausible strategy for continually adapting the representation network for a continual stream of data. Another interesting extension to the work would be to use the OML objective to meta-learn some other aspect of the learning process – such as a local learning rule (Metz et al., 2019) or an attention mechanism – by minimizing interference.

# 7 Acknowledgements

The authors would like to thank Hugo Larochelle, Nicolas Le Roux, and Chelsea Finn for helpful questions and feedback, and anonymous reviewers for useful comments.

## Footnotes

[1]Code accompanying paper available at https://github.com/khurramjaved96/mrcl

[2]This definition encompasses the continual learning problem where the tuples also include task descriptors $T_t$ (Lopez-Paz and Ranzato, 2017). $T_t$ in the tuple $(X_t, T_t, Y_t)$ can simply be considered as part of the inputs.

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
