[Supplementary Material]

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

Table 3: Parameters for Sinusoidal Regression Experiment

| Parameter | Description | Value |
|-----------|-------------|-------|
| Meta LR | Learning rate used for the meta-update | 1e-4 |
| Meta Update Optimizer | Optimizer used for the meta-update | Adam |
| Inner LR | LR used for the inner updates for meta-learning | 0.003 |
| Inner LR Search | Inner LRs tried before picking the best | [0.1, 1e-6] |
| Steps-per-function | Number of gradient updates for each of the ten tasks | 40 |
| Inner steps | Number of inner gradient steps | 400 |
| Total layers | Total layers in the fully connected NN | 9 |
| Layer Width | Number of neurons in each layer | 400 |
| Non-linearly | Non-linearly used | relu |
| RLN Layers | Number of layers used for learning representation | 6 |
| Pre-training set | Number of functions in the meta-training set | 400 |

Table 4: Parameters for Omniglot Representation Learning

| Parameter | Description | Value |
|-----------|-------------|-------|
| Meta LR | Learning rate used for the meta-update | 1e-4 |
| Meta update optimizer | Optimizer used for the meta-update | Adam |
| Inner LR | LR used for the inner updates for meta-learning | 0.03 |
| Inner LR Search | Inner LRs tried before picking the best | [0.1, 1e-6] |
| Inner steps | Number of inner gradient steps | 20 |
| Conv-layers | Total convolutional layers | 6 |
| FC Layers | Total fully connected layers | 2 |
| RLN | Layers in RLN | 6 |
| Kernel | Size of the convolutional kernel | 3x3 |
| Non-linearly | Non-linearly used | relu |
| Stride | Stride for convolution operation in each layer | [2,1,2,1,2,2] |
| # kernels | Number of convolution kernels in each layer | 256 each |
| Input | Dimension of the input image | 84 x 84 |

# Appendix

# A    Reproducing Results

We release our code, and pretrained OML models for Split-Omniglot and Incremental Sine Waves at https://github.com/Khurramjaved96/mrcl. In addition, we also provide details of hyper-parameters used from learning the representations of Incremental Sine Waves experiment and Split-Omniglot in Table 3 and 4 respectively.

For online learning experiments in Figure 4, we did a sweep over the only hyper-parameter, learning rate, in the list [0.3, 0.1, 0.03, 0.01, 0.003, 0.001, 0.0003, 0.0001, 0.00003, 0.00001] for each method on five validation trajectories and reported result for the best learning rate on 50 random trajectories.

## A.1    Computing Infrastructure

We learn all representations on a single V100 GPU; even with a deep neural network and meta-updates involving roll-outs of length up to 400, OML can learn representations in less than five hours for both the regression problem and omniglot experiments. For smaller roll-outs in Omniglot, it is possible to learn good representations with-in an hour. Note that this is despite the fact that we did not use batch-normalization layers or skip connections which are known to stabilize gradient based meta-learning.

Figure 7: Overview of a single gradient update for representation learning. (1) We adapt the PLN parameters by online updating $W_0$ to get $W_k$. (2) We compute the meta-loss on a batch of the complete data trajectory using updated parameters $W_k$. (3) Finally, we compute the gradients of the meta-loss w.r.t $\theta$ and update $\theta$ to $\theta'$. Optionally, we might also update $W_0$ to $W_0'$.

OML                    SR-NN (4.9%)

SR-NN (15%)            Pretraining

Figure 8: More samples of representations for random input images for different methods. Here SR-NN (4.9%) is trained to have similar sparsity as OML whereas SR-NN (15%) is trained to have the best performance on Split-Omniglot benchmark.

Figure 9: Instead of learning an encoder $\phi_\theta$ , we learn an initialization by updating both $\theta$ and $W$ in the inner loop of meta-training. In "OML without RLN," we also update both at meta-test time whereas in "OML without RLN at test time," we fix $theta$ at meta-test time just like we do for OML . For each of the methods, we report the training error during meta-testing. It's clear from the results that a model initialization is not an effective bias for incremental learning. Interestingly, "OML with RLN at test time" doesn't do very poorly. However, if we know we'll be fixing $\theta$ at meta-test time, it doesn't make sense to update it in the inner loop of meta-training (Since we'd want the inner loop setting to be as similar to meta-test setting as possible.

Figure 10: Average activation map for the best performing SR-NN with 15% sparsity. Scale goes from 0 to max (Light to dark green.)

## B  Representations

We present more samples of the learned representations in Figure 8. We also include the averaged representation for the best performing SR-NN model (15% instance sparsity) in Figure 10 which was excluded from Figure 6 due to lack of space.

---

**Algorithm 3:** Meta-Training : Approximate Implementation of the OML Objective

---

**Require:** $p(\mathcal{T})$: distribution over tasks
**Require:** $\alpha, \beta$: step size hyperparameters
**Require:** $m$: No of inner gradient steps per update before truncation
 1: randomly initialize $\theta, W$
 2: **while** not done **do**
 3:     Sample task $\mathcal{T}_i \sim p(\mathcal{T})$
 4:     Sample $\mathcal{S}^i_{train}$ from $p(\mathcal{S}_k|\mathcal{T}_i)$
 5:     $W_0 = W$
 6:     $\nabla_{accum} = \mathbf{0}$
 7:     **while** $j \leq |\mathcal{S}_{train}|$ **do**
 8:         **for** $k$ in $1, 2, \ldots, m$ **do**
 9:             $W_j = W_{j-1} - \alpha \nabla_{W_{j-1}} \ell_i(f_{\theta, W_{j-1}}(X^i_j), Y^i_j)$
10:             $j = j + 1$
11:         **end for**
12:         Sample $S^i_{test}$ from $p(\mathcal{S}_k|\mathcal{T}_i)$
13:         $\theta = \theta + \nabla_\theta \ell_i(f_{\theta, W_j}[S_{test}[0 : j, 0]], S^i_{test}[0 : j, 1])$
14:         Stop Gradients$(f_{\theta, W_j})$
15:     **end while**
16: **end while**

---

### B.1  Why Learn an Encoder Instead of an Initialization

We empirically found that learning an encoder results in significantly better performance than learning just an initialization as shown in Fig 9. Moreover, the meta-learning optimization problem is more well-behaved when learning an encoder (Less sensitive to hyper-parameters and converges faster). One explanation for this difference is that a global and greedy update algorithm – such as gradient descent – will greedily change the weights of the initial layers of the neural network with respect to current samples when learning on a highly correlated stream of data. Such changes in the initial layers will interfere with the past knowledge of the model. As a consequence, an initialization is not an effective inductive bias for incremental learning. When learning an encoder $\phi_\theta$, on the other hand, it is possible for the neural network to learn highly sparse representations which make the update less global (Since weights connecting to features that are zero remain unchanged).