[Reviews · NeurIPS 2019]

Reviewer 1



* Writing and clarity - Some descriptions are unclear and hard to follow, e.g., L125-126. - The abbreviation MRCL is used before its definition (Abstract, Fig.3, L171). - How do you define the marginal distribution \mu in L73? Please clearly define -- the number of samples for a given X_t? Then, it's not a probability distribution. - In section 2, the authors assumed "a variety of correlated sequences" (L74). However, the rationale of OML in Eqn.3 is not sufficient to support the "correlated sequences". Even though consulting with Appendix B, the rationale is weak to persuade. Do you assume that the k-step online update in Eqn.3 can give optimal loss for RLN although k is smaller than the length of each session? And, what do you mean by "finding a model initialization and learning a fixed representation such that starting from the learned representation it has xyz properties (Appendix L379-380)"? - I strongly recommend the authors to move Algorithm 1 from Appendix to the paper after polishing Section 2 & 3. * Related works Missing related works make it hard to assess the novelty and significance of the proposed method. If it is possible, please do report the controlled experiments to compare state-of-the-art. Some parts of the comparison with the other methods should be mentioned in the paper to shape the position among the related works. The following papers (not exhaustively listed) are recommended to consider: 1) Meta-learning with Latent Embedding Optimization, Rusu et al., 2018 2) TADAM: Task Dependent Adaptive Metric for Improved Few-shot Learning, Oreshkin et al., 2018 3) Task-Agnostic Meta-Learning for Few-shot Learning, Jamal et al., 2019 * Evaluations - L209-210 says that training accuracy is the indicator for the amount of forgetting. However, Fig.3 shows the limited aspects of analysis on the catastrophic forgetting and how the models persistently maintain the information. Could you plot the progressive results of the accuracies? - There are only two experiments using sine waves and Split-Omniglot, which are both too simple to confirm its significance. More complex and realistic datasets should be tested, e.g., CIFAR100 (Krizhevsky, 2009), Mini-Imagenet (Vinyals et al., 2016), and Tiered-ImageNet (Ren et al., 2018).

Reviewer 2



Originality: There are previous works that fixes the feature representation while only adapts the task learning part for new tasks. AFAIK there's no previous work using meta learning for this. Quality and Clarity: The idea is simple and well illustrated in the paper, however the experimental protocol is less clear. 1. Does the baseline pretraining mean that the model is trained with iid pretraining and then go over the training set again with online setting? The pretraining baseline having the same name as the pretraining set makes it a bit hard for the reader. (If I understand correctly, the pretraining set means the meta-training set.) 2. In split omniglot (a), is the random batch sampled from the entire pretraining set or just those that are already observed? (b) says sampling a single class, is the training phase a one pass through a trajectory? or does it allow revisiting of the data? It needs to the clarified. 3. Note that class id is implicitly used during omniglot training where a batch is sampled from a single class, which is in conflict with the claim that no task id is used. 4. I assume the first figure of Figure 4 is not very necessary because the training phase use a random batch for meta training, so it is implicitly an iid method, it is not very interesting to see its comparison to the other methods, could even be misleading that the MRCL is performing so close to Oracle. Limitations: 1. The evaluation with only Omniglot as the real dataset is limited, because Omniglot is known to show a favorable result for meta learning. 2. This method has a training phase and testing phase, in the training phase random sampling from previous data is required, thus the training phase is implicitly an IID method with cross validation. Only the test phase is under valid continual learning setting. This seems to assume that we could learn on a fixed training set and fixed the representation for all future distribution we encounter. This is not true, the reason why meta learning is successful is that the meta-training set should have a distribution that is similar to the meta-test set. This is incompatible with the continual learning goal that we want to dynamically integrate new knowledge into the network. Overall, I think the contribution of this paper is to use meta-learning (which basically means learning by cross validation to me) to find a representation that is potentially more disentangled and less prone to forgetting when adapting the prediction layer. This is a valid problem, but limited for continual learning.

Reviewer 3



Overall, this is a well-written paper that describes an interesting approach to continual learning (namely, meta-learning good initial representation to facilitate online learning). The paper draws connections to the literature well and provides interesting analysis of the representations learnt in this fashion. One shortcoming is that the only two datasets/tasks are considered and these are not large-scale enough to completely evaluate the performance of the method. There’s an open question of how this method will scale to larger tasks and the feasibility of this approach. For example, what do you pre-train on if you want to do continual learning on Atari games? Nonetheless it should provide a good starting point for further research. The rest of this review provides detailed comments as well clarifications that are required. This paper deals with the important issue of continual learning (CL). Broadly, the objective of continual learning is to allow a network to learn a series of tasks online, without forgetting previously learnt tasks (catastrophic forgetting). There are a range of different ways to tackle this problem, and this paper starts by providing a clear motivation and a comprehensive literature review. Most current methods are largery optimisation based, or memory/replay based. The approach (MRCL) this paper takes is to learn a representation such that when used in a CL setting, it allows for greater transfer and less interference. This is similar to a MAML-style method for meta-learning, where the idea is to find an initialization that allows for rapid adaptation. Instead here, we want a representation that allows for continual learning. (Another work to reference that is missing from the paper is “Memory-based parameter adaptation” that does a similar MAML style update albeit from a memory-buffer). The method here involves splitting the network into two sets of weights: a representation learning network (RLN, weights W) and a task learning network (TLN, weights T). The TLN will be trained to maximise reward/minimise loss on the task at hand, with fixed representation from the RLN. After doing this for k steps, we will have a set of weights W_k. We will then sample a fresh batch and update the initial W_0 and T_0 wrt to the loss on this current batch with parameters W_k and T_0, to give us W’ and T’. Thus, we meta-learn an initial representation that allows for good online learning. There is good intuition provided as to what such a representation looks like wrt the solution manifold. * Please clarify in equation (3) what the expectation is taken with respect to. Further please consider moving the diagram and pseudocode from the appendix to the main paper for clarity. The first task considered is an incremental sine wave task where there are K sine waves to be regressed. At pre-training, the RLN is meta-trained in the fashion above, and at the evaluation time is fixed as a feature extractor. This is compared to a pure pre-training method and another SR-NN method. Please provide more details (possibly in the appendix) for how this sparse SR-NN method works. We find the MRCL allows less interference than other method. This approach is also repeated for split-omniglot. The paper also provides good insight into sparsity that results from training in this fashion However, it would be helpful if the authors could provide a concrete definition of “instance sparsity”. This section could also be shortened somewhat. One question is how more traditional methods like EWC or memory-based methods would do in this regime. This method uses a modified version of the standard CL setup: ie allowing pre training before the tasks. This is evaluated in the last section where MRCL is viewed as orthogonal to these methods and combined as a pre-training method. Could the authors however please clarify the numbers in Table 1 -- are these average over all tasks? How do these related to figure 4? The numbers for EWC are surprisingly low, to me. Could you provide some intuition as to why? How converged were the models when the fisher matrix is calculated? One possible extension would also be to run this on permuted MNIST (where you pre-train on another set of permutations) and compare to doing standard EWC with/without pre-training. This would allow the numbers to be compared to more standard benchmark tasks in the literature. Overall, this paper is a very useful contribution to the field and will provide a significant starting point for future research. I recommend acceptance.

[Author Response · NeurIPS 2019]

We thank the reviewers for the valuable feedback. There were two common concerns: lack of a complex benchmark and unclear terminology in the experiments. We address these first and then follow with responses to each reviewer.

**Other Benchmarks:** We didn't run experiments on a more complex dataset because even on split-omniglot, existing continual learning methods perform extremely poorly. It is fair question if the conclusions extend to other datasets. We therefore ran our method on Mini-Imagenet and report the results in Figure 1. We incrementally learn a 20-way classifier using 30 samples per class for both train and test. The results support the same conclusions. Note that we go over the training trajectory *only once*, one sample at a time.

Figure 1: Reproducing the classification results on Mini-Imagenet.

We will include these results to provide further evidence for MRCL and that the strategy scales to more complex settings.

**To clarify the experiment protocol** we have decided to use updated terminology in the paper. Our method is divided into two phases : (1) meta-training phase and (2) meta-testing phase. The meta-training phase involves optimizing the OML objective for learning a representation and the meta-testing phase involves training the **TLN** on a highly correlated trajectory *in a single pass*. There is no overlap between data used in meta-training and meta-testing. All results are reported in the meta-testing phase; we also do not use IID sampling or multiple epochs for MRCL in any of the reported results. The first figure of Figure 4 is very meaningful because it shows MRCL—which learns incrementally in class order—is almost as effective as the Oracle—which learns using IID data. This highlights how much a reprsentation trained for online updating can help mitigate interference.

**Reviewer 1: Writing and clarity:** Thank you for pointing out issues with the writing and L73; we will fix those and move the algorithm to the main paper. In Appendix L379-380, we meant that a meta-learned initialization alone can not solve the interference problem; it is important to transform the input into a representation with non-interfering solution manifolds.

**Improvements: ... increase your score? (1)** The intuition behind MRCL is that instead of using sparsity as a proxy for good representations for continual learning, we directly measure interference caused by highly correlated updates over a finite horizon, and minimize this interference to learn a representation. We assume that a representation that minimizes interference for $k$ correlated updates would also reduce interference in the long run. In the incremental sine experiment, $k$ is actually equal to length of the complete trajectory whereas for Omniglot, $k$ is much smaller than the complete trajectory. Empirical results support that in both cases, OML can recover a good non-interfering representation. One explanation is that as long as $k$ is large enough to cause measurable interference, minimizing OML will result in a good representation.

**(3):** It's not clear how to compare our method with the three suggested approaches. One of the three approaches, TADAM, is specific to few-shot-learning - a different problem setting than ours. The remaining two approaches improve on gradient based meta-learning in general and **are complementary to our work** i.e. they can be combined with our objective function to potentially further improve the results. To better clarify this, we will extend the related work section of our paper to explain why a comparison with these approaches is tangential to our contributions.

**Reviewer 2: Quality and Clarity: (1)** Yes. The model is trained on the meta-training set using iid sampling, and online learning is done on meta-test set in a single pass. **(2)** Random batch is sampled from the entire meta-training set. Meta-training involves revisiting the data, whereas training during meta-tesing involves a single pass through data. **(3)** We agree that since class label is the same as class id, class id is implicitly available. However, our method does not exploit it in anyway, and learns a single classifier over all classes across task ids. We will nonetheless fix the inaccuracy in our claim. **(4)** The first figure in Figure 4 is the *training error during meta-testing* and does not involve IID sampling or multiple epochs. It measures degree of forgetting without taking into account the generalization error, and does in-fact perform very close to Oracle.

**Limitations: (2)** We fully agree that a fixed representation can not solve continual learning. We addressed this limitation in L273-L275 by suggesting one strategy which can be used to continuously update the representation. For the purpose of this paper, we focused on demonstrating that an effective representation can greatly reduce interference. We are currently extending this work using this proposed strategy, with a slowly changing representation updated using the OML objective.

**Reviewer 3: Please... sparse SR-NN method works:** SR-NN regularizes the activations across a mini-batch to be instance sparse where a feature is x% instance sparse if it is non-zero for x% of examples in a batch/mini-batch of data. We will add a more detailed description of SR-NN, and a precise definition of instance sparsity in the appendix.

**Equation (3)** The expectation is taken with respect to all possible length $k$ trajectories, starting from $X_t = x$: over all the random variables $\{(X_{t+i}, Y_{t+i})\}$. The outer integral is an expectation over $X_t$, according to distribution $\mu$.

**Diagram and pseudocode** We will move the pseudocode and diagram to the main paper.

**Could ... How do these related to Figure 4?** The results on the left of Table 1 (One class per task, 50 tasks) correspond to x=50 in Figure 4. Online + MRCL correspond to the MRCL line at x=50 whereas Online + Pretraining corresponds to Pretraining line at x=50.

**...EWC are surprisingly low .. why?** This is a great question. There are two reasons. (1) EWC tends to do extremely poorly on incremental classification tasks. (2) It does poorly when using a single pass through the trajectory, because the model does not necessarily converge on a task in a single pass. Our results are consistent with those reported by Riemer *et.al* 2019, Chaudhry et.al 2019 and others.

[Meta-Review · NeurIPS 2019]

Two of the reviewers increased their score after reading the rebuttal. All three reviewers now provide accepting scores. The reviewers particularly appreciated the authors response. In particular the additional experiment on mini-imagenet as it re-confirms the original idea and gives consistent results as those obtained with simpler datasets. The idea of borrowing meta-learning ideas to tackle the continual learning problem is interesting and the empirical results sufficient. The AC encourages the authors to include clarifications of the experimental protocol in the final manuscript and the discussion of why the performance of EWC is that low (as done in their response).